# Influence of COVID-19-Related Interventions on the Number of Inpatients with Acute Viral Respiratory Infections: Using Interrupted Time Series Analysis

**DOI:** 10.3390/ijerph20042808

**Published:** 2023-02-04

**Authors:** Jin-Won Noh, Woo-Ri Lee, Li-Hyun Kim, Jooyoung Cheon, Young Dae Kwon, Ki-Bong Yoo

**Affiliations:** 1Division of Health Administration, College of Software and Digital Healthcare Convergence, Yonsei University, Wonju 26493, Republic of Korea; 2Department of Health Administration, Graduate School, Yonsei University, Wonju 26493, Republic of Korea; 3Division of Cancer Control & Policy, National Cancer Control Institute, National Cancer Center, Goyang 10408, Republic of Korea; 4Department of Healthcare Institution Support, National Health Insurance Service, Wonju 26464, Republic of Korea; 5Department of Nursing Science, Sungshin Women’s University, Seoul 02844, Republic of Korea; 6Department of Humanities and Social Medicine, College of Medicine, The Catholic University of Korea, Seoul 06591, Republic of Korea; 7Catholic Institute for Healthcare Management, The Catholic University of Korea, Seoul 06591, Republic of Korea

**Keywords:** COVID-19, social distancing, acute respiratory infections, infectious disease, health policy, public health

## Abstract

After the first COVID-19 patient was diagnosed, non-pharmaceutical interventions such as social distancing and behavior change campaigns were implemented in South Korea. The social distancing policy restricted unnecessary gatherings and activities to prevent local transmission. This study aims to evaluate the effect of social distancing, a strategy for COVID-19 prevention, on the number of acute respiratory infection inpatients. This study used the number of hospitalized patients with acute respiratory infection from the Infectious Disease Portal of the Korea Centers for Disease Control and Prevention (KCDC) between the first week of January 2018, to the last week of January 2021. Intervention 1t represents the first patient occurrence of COVID-19, Intervention 2t represents the relaxing of the social distancing policy. We used acute respiratory infection statistics from Korea and segmented regression analysis was used. The analysis showed that the trend of the number of acute respiratory infection inpatients decreased after the implementation of the first patient incidence of COVID-19 due to prevention activities. After the relaxing of the social distancing policy, the number of inpatients with acute respiratory infections significantly increased. This study verified the effect of social distancing on the reduction in hospital admissions for acute respiratory viral infections.

## 1. Introduction

The coronavirus disease of the 2019 (COVID-19) pandemic was declared by the World Health Organization (WHO) on 11 March 2020, pointing to the over 118,000 diagnosed COVID-19 cases worldwide [1]. On 29 December 2019, several cases of pneumonia outbreak with unknown etiology were reported in Wuhan, China. Approximately seven days later, a gene sequence of COVID-19 revealed that the causative agent was a coronavirus, which was subsequently called SARS-CoV-2 [2]. According to the WHO Coronavirus Dashboard, as of 6:19 p.m. CEST, 27 September 2021, there have been 231,703,120 confirmed cases of COVID-19 and 4,746,620 deaths [3].

South Korea, as China’s neighboring country, could not fully take the precautions and preparation to this newly occurring COVID-19 infections until mid-February, 2020 [4]. However, multiple clusters of COVID-19 local transmission were identified in South Korea from February 2020. Various COVID-19 transmission and protective measurements, using extensive confirmed case screening, effective patient triage systems, and accurate information sharing have been introduced in South Korea [4].

After the first COVID-19 patient was diagnosed, non-pharmaceutical interventions such as social distancing and behavior change campaigns were implemented in South Korea [5,6]. The social distancing policy restricted unnecessary gatherings and activities to prevent local transmission [7]. Social distancing started voluntarily at first and then was mandated by the Korean government. It was not allowed to use public transportation or to do social activities without social distancing [8,9]. At the same time, the government instructed face masks to be worn and hands to be washed. In the absence of a safe and effective vaccine or pharmaceutical intervention, social distancing has been used to mitigate local virus transmission [10]. Social distancing has been an effective measure to mitigate the emergence and spread of COVID-19 and has reduced the burden on the healthcare system [6].

Social distancing has been known to complement other pandemic planning measures such as vaccination and antiviral stockpiling to reduce pandemic influenza transmission [11]. Previous studies discovered social distancing reduced infectious eye diseases [12] and the incidence of diseases such as influenza virus, acute otitis media, the common cold, bronchiolitis, croup, gastroenteritis, influenza, nonstreptococcal pharyngitis, pneumonia, sinusitis, skin and soft tissue infections (SSTIs), streptococcal pharyngitis, and urinary tract infection [13].

Public hygiene management such as mask wearing and using hand sanitizers would be effective to prevent disease transmission. However, only a few studies have investigated the association between COVID-19 intervention and acute respiratory infections [14,15]. It is crucial to understand the impact of nonpharmaceutical interventions on respiratory infection. Therefore, this study aimed to investigate the effect of social distancing, a strategy for COVID-19 prevention, on the number of acute respiratory infection (ARI) inpatients.

## 2. Materials and Methods

### 2.1. Data

This study used the number of hospitalized patients with ARIs from the Infectious Disease Portal of the Korea Centers for Disease Control and Prevention (KCDC) [16]. The KCDC Infectious Disease Portal provides weekly data on infectious diseases. The KCDC monitors and collects infectious disease data continuously and regularly from designated institutions [17].

The data included the number of patients hospitalized with adenovirus, human bocavirus, parainfluenza virus, respiratory syncytial virus, rhinovirus, and human metapneumovirus. The study period was from the first week of January 2018, to the last week of January 2021. Temperature and relative humidity data were obtained from the Korea Meteorological Administration. On 19 January 2020, the first COVID-19 patient was reported in South Korea. South Korea’s social distancing policy began on 22 March 2020 and lasted for 45 days. The social distancing policy restricted the operation of some facilities (religious, indoor sports, and entertainment) and recommended that people stay at home as much as possible. Then, the government relaxed the policy on 6 May 2020 [6]. In order to analyze the impact of COVID-19 policy on the number of patients hospitalized with ARIs caused by viruses, the periods were divided as follows:(1)Period 1: Week 1 of January 2018 to Week 3 of January 2020. (Before the first incident of COVID-19.)(2)Period 2: Week 4 of January 2020 to Week 4 of April 2020. (After the first incident of COVID-19 and during the implementation of the social distancing policy.)(3)Period 3: Week 1 of May 2020 to Week 4 of January 2021. (The relaxing of the social distancing policy.)

### 2.2. Study Variables

The dependent variables in this study were the total number of hospitalized ARI patients and the number of hospitalized ARI patients for each virus. The dependent variables were calculated on a weekly basis. Adenovirus, human bocavirus, parainfluenza virus, respiratory syncytial virus, rhinovirus, and human metapneumovirus were included in the analysis. The total number of hospitalized ARI patients was calculated by excluding patients with coronavirus from the total number of ARI inpatients.

Month, temperature, and humidity were included as covariates. We created 11 seasonal dummy variables to capture seasonality. Weekly average temperatures and relative humidity were calculated from the daily data. The number of ARI patients was affected by the weather [18,19,20,21].

### 2.3. Statistical Analysis

Segmented regression analysis of the interrupted time series was used to assess the effect of COVID-19 policies on the number of inpatients with ARI. Segmented regression is a quasi-experimental approach to evaluate the effects of intervention over time [22].
(1)$$Y_{t}=\beta_{0}+\beta_{1}{time}_{t}+\beta_{2}{intervention1}_{t}+\beta_{3}time\;after\;{intervention1}_{t}+\beta_{4}{intervention2}_{t}+\;\beta_{5}time\;after\;{intervention2}_{t}+\beta_{6}temperature+\beta_{7}humidity+\sum_{i=1}^{12}\beta_{mi}month\;i+e_{t}$$
where $Y_{t}$ represents the number of hospitalized patients with viral ARI, $time_{t}$ represents the baseline trend (continuous), $intervention1_{t}$ represents the first patient occurrence of COVID-19, $intervention2_{t}$ represents the relaxing of the social distancing policy, $time\;after\;intervention\;1,2_{t}$ represents the period after intervention (continuous), Month(Feb)-Month(Dec) are indicators of monthly dummy variables for seasonality, and $e_{t}$ is the error term.

The time variable represents the baseline trend. Intervention indicates when intervention occurred. Intervention 1 was 0 before the first occurrence of COVID-19, and 1 after the first occurrence. Intervention 2 was 0 before the relaxing of the social distancing policy and 1 after the relaxing of the social distancing policy.

The interpretation of segmented regression analysis is difficult because the effects of policy change over time. So, we calculated the marginal effects on the dependent variables. Marginal effects can be used to express how a dependent variable changes when a specific independent variable changes [23]. $\beta_{1}$ is the coefficient associated with the baseline time trend, and $\beta_{2}$, $\beta_{3}$, $\beta_{4}$, and $\beta_{5}$ are the coefficients associated with the effect of the policy.

The marginal effects of the first patient incidence of COVID-19 and the relaxing of the social distancing were calculated on the fourth week of April 2020, the first week of July 2020, and the first week of January 2021. The fourth week of April 2020 represented right before the relaxing of the social distancing policy. The first week of July 2020 was selected to calculate short-term effects, and the first week of January 2021 is for long-term effects. These marginal effects were calculated as the difference between two adjusted means of *Y* for each relevant level of $\beta_{2}$, $\beta_{3}$, $\beta_{4}$, $\beta_{5}$.

A generalized estimation equation (GEE) was used to conduct the segmented regression model with an AR (1) working correlation matrix option and Poisson distribution. All statistical analyses were performed in SAS version 9.4 (Cary, NC, USA. SAS Institute Inc.).

## 3. Results

Figure 1 represents the time trends of the hospitalization of ARI patients for each virus. Looking at the overall trend, we can see a constant pattern of rapid increase in inpatients during spring (March–May) and fall (September–November) and a decrease in summer (June–August) and winter (December–February). Although the number of ARI patients increased after March, the number of ARI patients declined after the first patient incidence of COVID-19. In addition, the number of patients increased after the social distancing policy relaxed.

A segmented regression model was implemented to check the effects of two events related to COVID-19 (Table 1). The analysis showed that the month dummy variables showed significant effects on the numbers of ARI inpatients. While Intervention 1 at the first patient incidence of COVID-19 was mostly insignificant, the coefficients of the time after event Intervention 1 were significantly decreased for all ARI inpatients. The coefficient of the total number of all ARI inpatients was −0.262 (*p* < 0.001). After the relaxing of the social distancing policy, the number of inpatients with adenovirus, human bocavirus, parainfluenza virus, rhinovirus, and human metapneumovirus significantly increased. The total number of ARI inpatients showed increasing trends (0.263; *p* < 0.001). Figure 2 shows that the predicted values in our model were very similar to the actual values.

The marginal effects on the dependent variables are presented in Table 2. After the first patient incidence of COVID-19, the total number of ARI inpatients decreased on the fourth week of April 2020. After the ending of the social distancing policy, the number of ARI inpatients somewhat increased. Considering the effects of the two events, the total number of ARI inpatients decreased.

## 4. Discussion

### 4.1. Finding

The current study was designed to explore how preventive interventions, such as the social distancing policy, affected the number of ARI inpatients. The study findings confirmed that there was a significant association between the social distancing policy and the number of ARI inpatients. In this study, a repetitive pattern of a rapid increase in ARI inpatients during the spring (March–May) was not seen in South Korea because the social distancing policy began on 22 March 2020. However, the number of ARI inpatients in summer 2020 began to increase after the relaxing of the social distancing policy, in contrast to the previous pattern of decrease in ARI patients during the summer. The study findings support previous research demonstrating that social distancing has an impact on reducing ARIs related to influenza, enterovirus, and pneumonia, as well as a reduction in COVID-19 patients [10,24,25,26,27].

### 4.2. Interpretations

In Korea, the social distancing policy restricts close, face-to-face interaction in public places. This policy was implemented with other public health interventions, such as wearing face masks, coughing etiquette, and hand hygiene. These study findings may support some benefits related to the social distancing policy rather than other public health interventions.

Although face masks are well known as a simple and safe measure to prevent foreign airborne sneeze and cough droplets, face masks do not completely prevent viral infection during close contact (<3–6 ft) [25,28,29]. A recent review regarding the transmission of SARS-CoV-2 found that the use of face masks had a large effect on the reduction in viral infection, but that the level of protection varied according to the type of face mask (e.g., N95, surgical masks) [25]. A Cochrane review reported that there was uncertainty about the effects of face masks, such as medical/surgical masks, regarding the reduction in respiratory viral infection compared to not wearing a mask [30]. However, two meta-analyses showed that viral transmission decreased and protection from the virus increased after physical distancing [25,31]. In addition, prior research shows that face masks are effective in preventing respiratory viruses when combined with social distancing [25]. Appropriate social distancing may protect people against symptomatic and asymptomatic disease by controlling the transmission of respiratory droplets from virus carriers [25,28,29]. Wearing facial masks is effective if combined with other NPIs, such as social distancing and avoidance of overcrowding.

Second, the social distancing policy was more acceptable than wearing face masks because the perception of wearing face masks, recommendations for mask usage by health authorities, and public compliance regarding wearing face masks vary from country to country [9,24,25,32,33]. Most Koreans (83.4–92.3%) reported practicing social distancing during the COVID-19 pandemic, which might indicate higher compliance than in western countries [9,32]. Low or inappropriate compliance with social distancing policy can reduce the effect of social distancing on viral transmission and can increase the risk of person-to-person viral transmission [10,28,29] Therefore, it is necessary to provide the general public with specific guidelines and strict regulations depending on the country (or culture), referring to examples from South Korea, Taiwan, and Hong Kong [9,24,26,32,34].

The Korean government recommended the postponement of the start of school in February, and most schools started online classes in April, which may have constrained social activities before the social distancing policy was implemented on 22 March. Postponing the start of school and school closures during February and March 2020 may explain the rapid decrease in ARI inpatients during spring (March–May) 2020, compared to the previous decreasing patterns in 2018 and 2019. These findings are consistent with those of previous studies that examined the effects of school closures on the transmission of COVID-19 and influenza infections [26,27,31]. This study could not adjust for the effect of school closures on respiratory infection because each school had a different starting period and a different time when they started online classes. Further studies may examine the effect of school closures and the start of in-school classes on the incidence of respiratory virus infections.

Interestingly, parainfluenza virus and human metapneumovirus infections decreased by the largest percentage in this study. Parainfluenza virus infection accounted for one-third to one-half of the total viral croup and was frequent in spring and summer before 2019 [34,35]. Human metapneumovirus infection was frequent in April and May 2018 and 2019, which was consistent with previous findings [34,35,36]. However, human metapneumovirus activity did not increase in April and May 2020. A previous study assumed that respiratory virus infection was associated with climate-specific factors such as temperature, relative humidity, diurnal temperature variation, and wind speed [36,37]; however, temperature and humidity had no significant effect on the number of patients with ARIs in this study. These findings indicate that social distancing might be a useful strategy for the prevention and control of seasonal respiratory viruses, especially these two viruses. Rhinovirus cases decreased by the lowest percentage among ARI inpatients in this study. This finding may be related to rhinovirus’s characteristic of having high resilience against environmental conditions and its long period of viral shedding from patients [34]. Therefore, early and strict social distancing with other public health interventions might be required for the control of rhinovirus infections.

This study verified the effect of social distancing on the reduction of hospital admissions for acute respiratory viral infections. Various quarantine rules were helpful, but the effect of social distancing was the most important and strongest in decreasing the number of ARI inpatients. In this study, the number of ARI inpatients in summer 2020 began to significantly increase after the relaxing of the social distancing policy even though other public health interventions, such as wearing face masks, coughing etiquette, and hand hygiene were still implemented. Previous studies have also provided evidence for the definitive effectiveness of social distancing on respiratory viral infection rather than other public health interventions [25,26,27,30,31]. This study provides evidence that there is a need for guidelines and information on social distancing measures that slow the spread of respiratory infection to improve compliance for the general public. The study findings might support policy decisions as governments and health authorities prepare to impose social distancing measures in future respiratory epidemics.

### 4.3. Limitations

However, this study has some limitations. First, we did not adjust for other public health interventions, such as face mask usage, hand washing, and school closures. Further empirical data would be required to help decide which public health interventions should be implemented or ended first and which combinations of social distancing policy should be implemented for respiratory pandemics. Second, public compliance regarding social distancing policy was not fully examined; therefore, more data regarding the compliance of the general public to social distancing are needed. Social distancing started voluntarily and was mandated by the Infectious Disease Control and Prevention Act [8]. These responses were evaluated as being successful [38]. The social distancing compliance rate in 2020 was 83–92% [9]. Although adherence was not evaluated in the model, our results can be seen as a result of the high compliance rate. Third, public knowledge, perceptions, and attitudes toward public health interventions are different in different countries. Further studies should consider these differences. Finally, this study could not reflect the entire incidence pattern of acute respiratory viral infections because the study only contained the data for hospital admissions for acute respiratory viral infections.

## 5. Conclusions

This study provides evidence that social distancing policy as a preventive intervention had a substantial impact on reducing the number of ARI inpatients in 2020 in South Korea. Additionally, this study provides evidence of the importance of non-pharmacological interventions for the prevention of respiratory infection as simple and inexpensive strategies. Social distancing policy may be more acceptable and have higher public compliance than other preventive measures. Therefore, early and strict implementation of social distancing policy with other public health interventions is needed in future respiratory epidemics.

## Figures and Tables

**Figure 1 ijerph-20-02808-f001:**
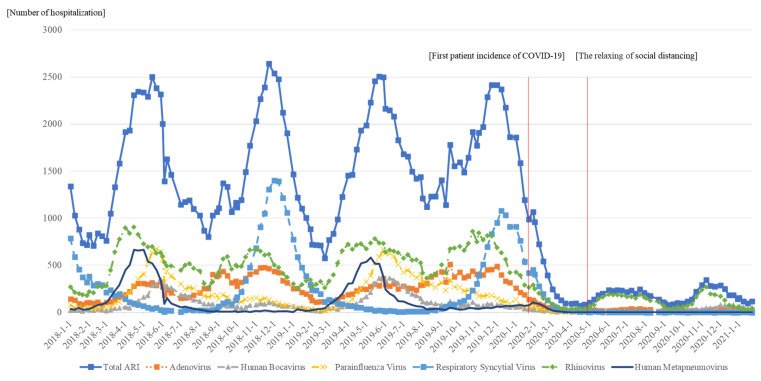
Time trends of hospitalization of acute respiratory infection inpatients.

**Figure 2 ijerph-20-02808-f002:**
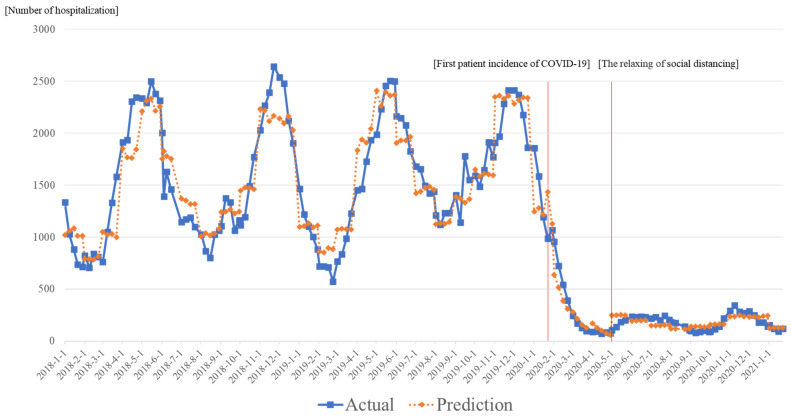
Total actual hospitalization of acute respiratory infection patients and predicted time trends.

**Table 1 ijerph-20-02808-t001:** Results of segmented regression analysis of the effect of COVID-19 on the number of inpatients with acute respiratory infection.

	Total Acute Respiratory Infection	Adenovirus	Human Bocavirus	Parainfluenza Virus	Respiratory Syncytial Virus	Rhinovirus	Human Metapneumovirus
Time	0.002 **	0.003 ***	0.003 ***	0.004 ***	0	0.003 ***	0.001
Intervention 1	0.398	0.031	0.252	0.341	0.23	0.3	1.787 ***
Time after i = Intervention 1	−0.262 ***	−0.198 ***	−0.208 ***	−0.32 ***	−0.158 ***	−0.278 ***	−0.506 ***
Intervention 2	1.174 ***	0.209	−0.972 ***	−0.525	0.66	2.231 ***	−0.086
Time after i = Intervention 2	0.263 ***	0.196 ***	0.315 ***	0.342 ***	0.017	0.264 ***	0.545 ***
Temperature	−0.001	−0.007	0.024	0.022	−0.074 ***	0.015	0.019
Humidity	0.003	0.004	−0.003	0.01 **	0.021 ***	0	0.005
February	−0.252 *	−0.353 ***	−0.276 *	−0.248	−0.419 **	0.046	0.612 ***
March	−0.056	−0.207	−0.045	−0.08	−0.781 ***	0.487 **	1.397 ***
April	0.525 ***	0.357 **	0.581 *	1.06 ***	−1.014 ***	0.882 ***	2.392 ***
May	0.724 ***	0.637 ***	1.595 ***	1.802 ***	−1.495 ***	0.636 ***	2.317 ***
June	0.473 *	0.46 **	1.663 ***	1.565 ***	−2.216 ***	0.453 *	1.079 *
July	0.166	0.333	1.058 **	1.01 **	−2.115 ***	0.283	0.097
August	−0.088	0.683 **	0.359	0.616	−1.651 **	−0.103	−0.826
September	0.081	0.813 ***	0.313	0.623 *	−0.716	0.256	−0.949 *
October	0.253	0.738 ***	0.316	0.635 **	−0.086	0.573 ***	−0.704 *
November	0.643 ***	0.877 ***	0.61 ***	0.657 ***	0.61 **	0.827 ***	−0.354
December	0.632 ***	0.715 ***	0.626 ***	0.4 *	0.577 ***	0.51 ***	0.001

*p* < 0.05 = *, *p* < 0.01 = **, *p* < 0.001 = ***.

**Table 2 ijerph-20-02808-t002:** Marginal effects of the first patient incidence of COVID-19 and the relaxing of the social distancing policy on the number of inpatients with acute respiratory infection.

Acute Respiratory Infection	The Effects on Fourth Week of April 2020 (Compared to Third Week of January 2020)	The Effects on First Week of July 2020 (Compared to Third Week of January 2020)	The Effects on First Week of January 2021 (Compared to Third Week of January 2020)
	Difference (95% CI)	Difference (95% CI)	Difference (95% CI)
Total acute respiratory infection	−1394.0 (−1493.7–−1294.3)	−1300.1 (−1386.9–−1213.3)	−1298.3 (−1376.4–−1220.4)
Adenovirus	−225.9 (−243.8–−208.1)	−223.4 (−241.3–−205.5)	−224.5 (−242.3–−206.7)
Human bocavirus	−88.2 (−96.0–−80.3)	−88.3 (−95.8–−80.7)	32.5 (5.3–59.8)
Parainfluenza virus	−175.1 (−194.8–−155.3)	−175.6 (−195.3–−156.0)	−174.3 (−193.6–−155.0)
Respiratory syncytial virus	−205.5 (−242.6–−168.4)	−217.9 (−255.3–−180.6)	−232.7 (−275.1–−190.3)
Rhinovirus	−449.9 (−482.0–−417.8)	−369.3 (−399.0–−339.5)	−389.2 (−418.7–−359.7)
Human metapneumovirus	−125.8 (−147.7–−104.0)	−125.7 (−147.4–−104.1)	−124.6 (−145.8–−103.4)

CI: confidence interval.

## Data Availability

The data is publicly available in the Infectious Disease Portal of the Korea Centers for Disease Control and Prevention and can be used after application through the link below. [https://www.kdca.go.kr/npt/biz/npp/iss/influenzaStatisticsMain.do, accessed on 27 January 2023].

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
