# Peer review of "Influence of COVID-19-Related Interventions on the Number of Inpatients with Acute Viral Respiratory Infections: Using Interrupted Time Series Analysis"

_ijerph, 2023, doi:10.3390/ijerph20042808_

Round 1

Reviewer 1 Report

Thank you for the opportunity to review the manuscript “Influence of COVID-19-related interventions on the number of inpatients with acute viral respiratory infections: Using interrupted time series analysis” (ijerph-2173723).

The study aimed to investigate the effect of social distancing as prevention on the number of acute respiratory infection inpatients.

The topic is interesting, but the paper needs to be fundamentally revised. Also a linguistic revision is necessary even in style (for example Line 197 be-cause, Line 203 point missing…).

Introduction: The authors should add further information about other pandemics and social distancing in relation to diseases. Also, Hypotheses should be formulated more clearly.

Discussion: Please summarize the main results of the study in a table. It is important that the introduction is better linked to the discussion.

Author Response

Dear Reviewer.

Thank you for your valuable comments.

We have written a point-by-point answer in response to your comment.

Please confirm the attached file.

Best regards.

WR Lee.

Reviewer 2 Report

1.Title

- I suggest putting in the title the intervention used that are “non-pharmaceutical interventions”.

2. Introduction

- Presents the contextualization of the theme in a logical sequence.

- The objectives are well formulated.

- In the last paragraph of the introduction, the section “However, only a few studies have investigated the association between COVID-19 intervention and acute respiratory infections”, include the references of the studies.

- Explain the knowledge gap based on the scientific literature and how your work will impact the knowledge gap.

3. Method

- The authors describe the type of study, the selection and composition of the sample/research participants, period and data collection process, the instruments used, the method used for data analysis.

- I suggest adding a description of how the social distancing policy was implemented.

- Highlight that adherence to the intervention was not evaluated.

- Describe whether any incentives were carried out to increase adherence to social distancing.

5. Discussion

- Since these are seasonal respiratory viruses, I suggest that the authors address the forms of transmission of the evaluated viruses, as well as reflect that non-pharmacological interventions in combination are more effective than using only one.

- Another very relevant point that the authors can deepen in the discussion is about adherence to the intervention, as the intervention has satisfactory results that are directly proportional to the adherence of the target audience, that is, presenting strategies to promote the intervention are essential.

- In the “Second, the social distancing policy was more acceptable than wearing face masks…”, justify the government's use of this intervention over other interventions by combining non-pharmacological interventions.

6. Limitations

- The limitations are very present and consistent with the study.

7. Conclusions

- The objectives of the study are clear and responsive, however, I think it is important for the authors to present a reflection on the importance of non-pharmacological interventions for the prevention of respiratory infections; the need to encourage their adherence; the choice of intervention or interventions, considering the space that will be implemented; the relevance of health education in the community; and paying attention to the clinical relevance of the study, as we are talking about lives that can be saved with simple and low-cost strategies.

- Studies that evaluate the effectiveness of non-pharmacological interventions in the prevention of respiratory infections are relevant, especially when they must be implemented and discontinued, according to the reality of each location. After the COVD-19 pandemic, we scientists need to instruct society about a change in behavior regarding respiratory etiquette, hand hygiene, social distancing, among other non-pharmacological interventions that are easy to perform and low cost and can save lives. Congratulations to the authors for conducting the study.

Author Response

(The authors gave the same response as above.)

Reviewer 3 Report

The authors aimed to investigate the effect of social distancing, a strategy for COVID-19 prevention, on the number of acute respiratory infection (ARI) inpatients.

The study sounds good, and the authors used accurate statistical tools to reach their objectives.

However, I would like to ask why the authors did not include other important pathogens causing hospital admissions such as Influenza virus and bacterial pathogens (e.g., Streptococcus pneumoniae and Haemophilus influenzae).

Moreover, I suggest increasing the study period until January 2021. It is obvious that the highest number of hospitalizations is between October and January. Including that period will make the conclusions stronger and more significant.

Author Response

(The authors gave the same response as above.)

Round 2

Reviewer 1 Report

The authors improved their paper.

Reviewer 3 Report

The manuscript can be accepted in its current form.